# Interventional Treatment of Complex Regional Pain Syndrome

**DOI:** 10.3390/biomedicines11082263

**Published:** 2023-08-14

**Authors:** Lamberta Ghaly, Vincent Bargnes, Sadiq Rahman, George-Abraam Tawfik, Sergio Bergese, William Caldwell

**Affiliations:** Stony Brook University Hospital, Stony Brook, NY 11794, USA; lamberta.ghaly@stonybrookmedicine.edu (L.G.); vincent.bargnes@stonybrookmedicine.edu (V.B.); sadiq.rahman@stonybrookmedicine.edu (S.R.); george-abraam.tawfik@stonybrookmedicine.edu (G.-A.T.); william.caldwell@stonybrookmedicine.edu (W.C.)

**Keywords:** complex regional pain syndrome, chronic regional pain syndrome, neuropathic pain

## Abstract

Complex regional pain syndrome (CRPS) is a rare but debilitating chronic pain disorder characterized by persistent pain disproportionate to any preceding injury. CRPS can have a significant impact on a person’s quality of life, often leading to disability and psychological distress. Despite being recognized for over a century, finding the right treatment for CRPS can be challenging. In this article, we will explore the causes, symptoms, and interventional treatment options for CRPS, as well as the latest research on this complex and often misunderstood condition.

## 1. Introduction

Complex regional pain syndrome (CRPS) is a rare chronic pain condition known by many names over the years, including reflex sympathetic dystrophy, causalgia, Sudeck’s atrophy, and algodystrophy. It is characterized by extreme pain and sensitivity disproportionate to a preceding injury and is not restricted to the distribution of a specific peripheral nerve. Along with the dominating feature of intense pain, CRPS encompasses a variety of clinical changes including motor, autonomic, and trophic changes, which can have a significant impact on a person’s quality of life, often leading to disability and psychological distress [1]. Despite being recognized for over a century, diagnosing and treating this perplexing syndrome remains a challenge.

The pathophysiology of CRPS remains obscure despite centuries of studies and attempts to understand the underlying process. In 1900, Paul Sudeck presented findings of bone atrophy that occurred after acute inflammation, ligament injury, soft tissue infection, or nerve injury. Sudeck stated that usually this atrophy disappeared quickly with full recovery of the patient; however, in some instances, the atrophy persisted and resulted in disability for the patient. This bone atrophy that remained chronically was later called Sudeck Atrophy [2]. In 1917, Rene Leriche was the first to find that the sympathetic nervous system is involved in symptoms of CRPS. In a patient with chronic hand pain and numbness after a gunshot wound, Leriche performed a sympathectomy and found that the patient reported full resolution of pain after the procedure. To describe the role of the sympathetic nervous system in neuropathic pain, Leriche created the name sympathetic neuritis [2]. In 1947, James Evans created the term Reflex Sympathetic Dystrophy. He theorized that tissue injury leading to afferent input would create a reflex arc with the spinal cord, resulting in the stimulation of sympathetic efferent neurons, ultimately resulting in dystrophy [3]. The current nomenclature of CRPS originated from a consensus conference organized in 1993 to review nomenclature and diagnostic criteria. The nomenclature Complex Regional Pain Syndrome was developed in 1994 by the International Association for the Study of Pain (IASP) to highlight the diverse pathogenesis of the disease and emphasize that the disease is localized to an anatomical regional rather than limited to the distribution of nerves or nerve roots [2].

The mechanism of CRPS is multifactorial and includes inflammatory, immunologic and brain plasticity changes. CRPS is involved in signs of inflammation such as increased temperature, swelling, redness and pain [4]. Elevated inflammatory markers have been found to play a role in CRPS, with studies showing high levels of pro-inflammatory cytokines such as TNF-alpha, Interleukin-1b, Interleukin-2, and Interleukin-6 in the serum and cerebrospinal fluid of patients [5]. These pro-inflammatory cytokines, along with calcitonin gene-related peptide, bradykinin and substance P, lead to vasodilation and plasma extravasation [4]. In terms of immunological changes, autoantibodies against beta-2-adrenergic receptor and muscarinic-2-receptor have been found to be involved in CRPS [6]. Central sensitization also plays a role in the pathogenesis of CRPS and contributes to the hyperalgesia and allodynia experienced in CRPS [4]. Central sensitization results from persistent noxious stimuli of peripheral nociceptive neurons [4]. This persistent noxious stimulation alters nociceptive processing in the central nervous system and results in increased excitability of central nociceptive neurons within the spinal cord [4]. The sensitization and increased activity of local peripheral and central nociceptive neurons is mediated by substance P, bradykinin and glutamate released by peripheral nerves [4]. Furthermore, CRPS is involved with cortical reorganization in the primary somatosensory cortex [7]. The affected extremity in CRPS was found to undergo shrinkage in the area of somatosensory–cortical representation compared to the unaffected extremity [7]. The cortical reorganization correlated with the amount of pain in CRPS and the level of hyperalgesia [7].

The diagnosis of CRPS is typically based on clinical signs and symptoms. There are two commonly used sets of criteria: the Budapest criteria and the IASP criteria. The Budapest criteria, established in 2003, include five categories of symptoms, including continuing pain disproportionate to any inciting event, sensory changes such as allodynia or hyperalgesia, vasomotor changes such as temperature asymmetry or skin color changes, sudomotor/edema changes such as edema or sweating changes, and motor/trophic changes such as decreased range of motion or muscle atrophy [8]. To meet the Budapest criteria, a patient must have at least one symptom in three of the four categories. The IASP criteria, established in 1994, include the presence of an initiating event, continuing pain, allodynia, or hyperalgesia, evidence of edema, changes in skin blood flow, or abnormal sudomotor activity in the region of pain, and no other condition that would account for the degree of pain and dysfunction [9]. To meet the IASP criteria, a patient must have all four criteria. While diagnosis can be challenging, healthcare professionals may use additional tests to help confirm the diagnosis, such as imaging studies or nerve conduction studies. The diagnosis of CRPS can be further classified as Type I, formerly known as reflex sympathetic dystrophy, or Type II, formerly known as causalgia. While both types share similarities, the key distinguishing feature is the presence of a distinct nerve injury, which is absent in CRPS-I and present in CRPS-II [10]. Studies have suggested that signs and symptoms of CRPS can also be clustered into subgroups based on sensory, vasomotor, and sudomotor dysfunction, which may provide a potential benefit in targeting treatment more effectively [11].

## 2. Multidisciplinary Approaches to Treatment

Due to the complexity of chronic regional pain syndrome, a multidisciplinary approach to treatment is often recommended. This involves a team of healthcare professionals from different disciplines working together to manage the various aspects of the condition. The team may include a pain specialist, a physiotherapist, an occupational therapist, a psychologist, and a social worker. Treatment may include a combination of medications, physical therapy, cognitive behavioral therapy, and interventional procedures. The goal of the multidisciplinary approach is to improve pain relief, physical function, and psychological well-being, and to help the patient regain their quality of life. While there is no cure for CRPS, a multidisciplinary approach can provide a comprehensive approach to managing the condition.

Research has explored the multifaceted nature of treating CRPS and proposed the integration of psychological approaches alongside medical and physical therapy (PT) to enhance management [12]. Two randomized controlled trials investigating PT’s efficacy for CRPS have integrated psychological components into the therapeutic regimens. Oerlemans et al., conducted a study implementing a PT protocol using relaxation exercises and cognitive interventions, aiming to empower patients to perceive greater control over their pain. This combined intervention yielded notably superior outcomes in terms of pain reduction, improved active range of motion, and reduced impairment levels compared to a social work control group [13]. Likewise, Lee et al., undertook an RCT for child and adolescent CRPS patients, wherein both PT groups received six cognitive behavioral treatment sessions in addition to their PT sessions. Children from 8 to 17 years of age (*n* = 28) were randomly assigned to either receive PT once per week for 6 weeks or PT 3 times per week for 6 weeks. Though there was no control for comparison, both PT groups exhibited significant enhancements in pain management and functional outcomes compared to their baseline measurements [14].

In addition to traditional physical therapy, treatments such as mirror therapy have been found to be successful in patients with CRPS. Mirror therapy, also known as graded motor imagery (GMI), is a non-invasive rehabilitation technique used for the treatment of CRPS involving the use of a mirror to create visual illusions that can help reduce pain and improve limb function in affected individuals. The evidence supporting the use of mirror therapy for CRPS treatment is based on various clinical studies and case reports. Several research studies have shown promising results in terms of pain reduction and functional improvements. While limited by small sample sizes and varying study designs, the overall findings suggest positive effects. In a randomized controlled trial, Moseley et al., found that mirror therapy was effective in reducing pain and improving movement in individuals with CRPS. The study involved 13 participants with CRPS, and those who received mirror therapy demonstrated significant reductions in pain intensity and improved motor function compared to a control group [15]. Moseley proposes that the characteristics of CRPS suggest a disparity between sensory input and central representation in the brain. In this context, he suggests that the application of mirror therapy has the potential to “rectify this dynamic central mismatch” [15]. In 2021, Strauss et al., investigated the effects of GMI on patients with CRPS lasting over 6 months. GMI was applied in 21 patients over 6 weeks to relieve movement pain of the upper limb. During the graded motor imagery intervention, movement pain was found to be decreased [16]. Moreover, pathological parameters such as increased activation in the primary somatosensory cortex during fist movement and decreased short intracortical inhibition were modified in the same way as movement pain and hand performance improved [16]. However, these changes were not observed during the waiting period when participants did not undergo the graded motor imagery intervention, demonstrating a benefit of mirror therapy.

In combination with physical and psychological therapy, medical management of CRPS involves various treatments targeting specific aspects of the condition. Neuropathic pain is a common feature of CRPS, and anticonvulsants are frequently used to address it. By modulating voltage-gated calcium channels in nerve cells, these medications stabilize hyperactive nerves and reduce the release of excitatory neurotransmitters, providing relief from neuropathic pain. Gabapentin, a first-line treatment for neuropathic pain, gained attention among pain specialists due to anecdotal reports of its effectiveness in CRPS [12]. It functions at the alpha 2-delta auxiliary subunit of voltage-dependent calcium channels, and well-designed large randomized controlled trials have demonstrated its efficacy in postherpetic neuralgia and diabetic peripheral neuropathy. While there is evidence from case series suggesting its efficacy in CRPS, gabapentin is widely empirically used for various neuropathic pain syndromes. Pregabalin, a closely related drug with the same mode of action, can be used as well, although there are currently no data evaluating pregabalin specifically for CRPS [12]. In cases of severe and refractory pain, opioids may be prescribed, acting on central nervous system receptors to block pain signaling and offer potent analgesia. However, their use requires caution due to harmful side effects, tolerance, and dependence, leading to reserved short-term or carefully selected use. Unlike acute nociceptive pain, neuropathic pain does not consistently respond to treatment, meaning that dose escalations are common. However, increasing the dosage often does not result in additional pain relief and may instead lead to the accumulation of adverse effects [12].

For localized pain management, topical medications can be applied directly to the affected area, minimizing systemic effects while providing targeted relief. Among the options available are lidocaine, capsaicin, and dimethyl sulfoxide (DMSO). Lidocaine, a potent local anesthetic, exerts its therapeutic action by selectively blocking voltage-gated sodium channels on nerve fibers in the application area [11]. By inhibiting the influx of sodium ions, lidocaine effectively interrupts nerve signal transmission, leading to temporary numbness and pain relief without significant systemic effects. Capsaicin, derived from chili peppers, depletes substance P, a neurotransmitter involved in pain transmission, leading to desensitization of pain receptors and providing pain relief [12]. However, it is important to follow proper application instructions to prevent potential skin irritation or adverse reactions. DMSO serves as an effective free radical-scavenging agent, demonstrating its potential as a therapeutic option. A systematic review reported promising results from a study involving the application of a 50% DMSO cream over a two-month period. This treatment approach demonstrated significant pain reduction compared to placebo, highlighting DMSO’s potential as an adjunctive therapy for pain management [12]. These findings contribute to the growing body of evidence supporting the utilization of topical medication as part of a comprehensive pain relief strategy, warranting further investigation and consideration.

Ketamine, an NMDA receptor antagonist, has also shown promise in providing rapid and profound pain relief in CRPS through controlled infusions that block pain signal amplification and central sensitization [17]. However, its administration requires cautious monitoring and patient selection due to potential side effects including dizziness, nausea, and increased sympathetic activation. Some patients may experience psychomimetic effects, such as hallucinations and dissociation, which can be distressing. Therefore, the risk-to-benefit ratio must be carefully assessed when considering ketamine therapy for CRPS patients. While these treatments play a crucial role in addressing the symptoms of CRPS and improving overall function, it is essential to acknowledge that, for individuals with persistent and refractory CRPS, medical treatments alone may have limitations in achieving complete pain relief. In such cases, interventional treatments can offer a promising pathway for further pain control and functional improvements. Collaborative discussions between patients and their healthcare providers can help determine the most appropriate and individualized treatment plan to effectively address the challenges posed by persistent CRPS.

## 3. Intravenous Regional Blockade

Intravenous regional blocks (IVRB) are one of many interventions shown to have benefits in treating CRPS. This procedure involves injecting local anesthetic into a limb to provide pain relief. Also known as a Bier block, it is a relatively quick and minimally invasive technique that can be performed in an outpatient setting. It involves placing a tourniquet on the affected limb followed by the injection of a local anesthetic, allowing the anesthetic to diffuse into the surrounding tissues and provide pain relief while reducing inflammation. IVRB is generally well-tolerated, and patients can typically return to their daily activities shortly after the procedure. Studies have shown benefits when used alone or in combination with other treatments for CRPS. 

In 1992, a study by Hord compared the effectiveness of intravenous regional block (IVRB) with bretylium to a lidocaine control in 12 patients. In a cross-over design, each patient received two treatments of bretylium + lidocaine and two treatments of lidocaine alone, with the duration of ≥30% pain relief being the measured outcome [18]. Results showed that IVRB patients experienced a significantly longer period of ≥30% pain relief (group mean duration 20.0 days) compared to the control group (group mean duration 2.7 days). However, the study had limitations including poorly defined diagnostic criteria and a high proportion of dropouts (5/12) with no intention-to-treat analysis [18]. 

Later in 1995, Connelly, Reuben, and Brull aimed to evaluate the effectiveness of IVRB using ketorolac and lidocaine in patients with clinical symptoms of sympathetically mediated pain, such as allodynia, hyperalgesia, hyperpathia, and edema. IVRB was offered to patients up to six times, with a maximum of one treatment per week for six weeks. The success of the blocks was measured based on the complete resolution of pain without requiring any other interventions other than the IVRA block, partial resolution of pain that was not long-lasting, or no subjective improvement after the IVRA block [19]. The study found that 26% of patients had a complete response to IVRA, 43% had a partial response, and 31% had no response. The study was limited by uncertainty regarding whether the relief was from ketorolac, lidocaine, or a combination of both, as well as the arbitrary dose of ketorolac that was chosen [19]. 

In 2010, Nascimento, Klamt and Prado compared the efficacy of IVRB versus stellate ganglion block (SGB) for the management of upper-extremity CRPS type 1. Fourteen patients were in the IVRB group and received a combination of 70 mg lidocaine and 30 µg clonidine. Patients receiving SGB were split into 2 groups: 14 patients received a SGB with 70 mg lidocaine alone, and 15 patients received a SGB with 70 mg lidocaine and 30 µg clonidine [20]. Each procedure was repeated at 7-day intervals a total of 5 times, and a visual analog scale (VAS) was used immediately before each procedure to measure pain intensity and duration. The study found that all groups had a significant reduction in pain scores on the VAS with a progressive reduction in pain from the first to the third block; the remaining fourth and fifth blocks did not produce a further decrease in pain in any of the groups [20]. The three groups had a comparable reduction in pain and did not differ significantly in terms of pain reduction as reported in the VAS. Drowsiness occurred most frequently in patients receiving SGB with lidocaine and clonidine and dry mouth occurred only in the SGB with lidocaine and clonidine group. 

Overall, IVRB holds promise as a valuable treatment option for CRPS, providing notable pain relief and facilitating improved quality of life for patients. Further research addressing the limitations of previous studies and exploring the optimal administration techniques and drug combinations can contribute to the refinement and wider adoption of IVRB in clinical practice.

## 4. Regional Sympathetic Nerve Blockade

Regional sympathetic nerve blocks involve injecting a local anesthetic under fluoroscopic or ultrasound guidance to block the activity of a sympathetic ganglion to provide pain relief. The stellate ganglion is blocked for patients with CRPS of the upper extremity and the lumbar sympathetic ganglion is blocked for patients with CRPS of the lower extremity. 

In 2009, Yucel et al., evaluated the use of stellate ganglion blockade (SGB) to treat CRPS type 1 in the hand. The 22 patients in the study received SGB with 15 mL of equal parts bupivacaine (5 mg/mL) and 1% prilocaine–hydrochloride (20 mg/mL). SGB was performed 3 times with a 1-week interval between treatments. The study measured pain intensity using a visual analog scale (VAS) before the initiation of the treatments and 2 weeks after the last SGB [21]. The range of motion of the wrist joint was also measured before and after treatment. The study found a significant decrease in pain reported on the VAS and a significant improvement in range of motion values for wrist flexion, extension, supination, and pronation However, these findings were limited by a small sample size and a lack of long-term blockade results. 

A retrospective study by Cheng et al., in 2019 investigated the therapeutic benefit of sympathetic blocks for the management of CRPS and the association between pain reduction and the pre-procedure temperature difference between the CRPS-affected limb and contralateral limb. The study investigated patients from 2009 to 2016 in a major academic center who received sympathetic blocks. Of the 255 observed patients with CRPS, 61% of them had successful pain reduction (pain reduction by >50%) after sympathetic block [22]. Out of the patients that did experience successful pain reduction, 81% of them had pain relief for 1–4 weeks or longer. It was found that there was no association between pre-procedure temperature of the limb and pain relief from the sympathetic blocks. The study was limited by the lack of a control group, nonblinding, and technical variations in the performance of the blocks [22].

More recently, in 2022, a randomized, double-blinded controlled trial by Yoo et al., investigated whether botulinum toxin would prolong the duration of lumbar sympathetic block through a sustained increase in skin temperature. The 48 participants in the study had unilateral CRPS of the lower extremity and were assigned to the experimental group receiving 75 IU of botulinum toxin type A or the control group receiving 0.25% levobupivacaine. The study found that the botulinum toxin group had a greater temperature increase from baseline than the control group [23]. This temperature increase in the affected extremity remained in the botulinum toxin group at 3 months but not in the control group. The botulinum toxin group also had a significantly greater decrease in pain at 1 month and 3 months post-procedure compared to the control group [23]. The study is limited by its small-scale, short follow-up duration of 3 months, and lack of measurement of inflammatory cytokines or electrophysiologic tests to verify changes in sensory symptoms. 

## 5. Spinal Cord Stimulation

Spinal cord stimulation (SCS) is recommended when other treatments have failed to improve CRPS pain and dysfunction. SCS involves implanting a small device under the skin that delivers electrical impulses to the spinal cord, with the goal of interrupting pain signals before they reach the brain, thus reducing pain sensations. SCS is believed to induce pain relief by activating Aβ fibers in the dorsal column, thereby producing varying effects on sensory perception and pain thresholds. This hypothesis can be traced back to the pioneering work of Wall and Melzack, who developed the gate control theory in 1965. The success of SCS in treating CRPS varies from person to person, but studies have shown that it can significantly reduce pain and improve quality of life for many patients.

Kemler et al., conducted a randomized study in 2000 involving 54 patients with refractory CRPS and found that SCS combined with standardized physiotherapy was more effective in reducing pain scores at 6 months compared to physiotherapy alone [24]. In the intention-to-treat analysis, the researchers observed an average pain reduction of 2.4 cm on the visual analog scale (VAS) at 6 months. However, for those who did receive spinal cord stimulation (SCS) treatment, the reduction was even greater, with a mean decrease of 3.6 cm [25]. In contrast, the control group experienced a slight increase of 0.2 cm in pain scores at the 6-month mark. Follow-up at 5 years showed a slight gradual decline in pain relief for SCS patients, as the mean VAS score exhibited a decrease of 2.5 cm from the baseline. On the other hand, the control group, which received physical therapy, displayed a 1 cm decrease at the 5-year mark (*p* = 0.06) [25].

In 2005, Harke et al., assessed the effect of SCS on functional status in 29 patients with CRPS-I in a prospective observational study. Pain intensity was recorded by patients using a VAS ranging from 0 (no pain) to 10 (unbearable pain). Patients’ functional impairment was rated using the pain disability index (PDI), with scores ranging from 0 (no disability) to 10 (total disability). After beginning SCS treatment, deep burning pain decreased from 10 to 2 on VAS and allodynia decreased from 10 to 0 on VAS, with results being similar at the 3-, 6-, 9-, and 12-month evaluations [26]. Functional impairment in daily activities was reduced after SCS as indicated by a decreased PDI of more than 50% [26]. During inactivation tests of SCS, reoccurrence of pain up to 8 VAS was measured 45 min after inactivation; there was also a skin temperature decrease of 1.5 °C in the affected limb compared to the contralateral unaffected limb [26]. After a mean follow-up period of 35.6 months, the median VAS for deep pain was still reported at 2 and allodynia was eliminated [26]. Additionally, after the follow-up period, 12 of 16 patients with CRPS of the upper limb showed significantly increased grip strength and 8 of 10 patients with lower limb disability resumed walking without crutches [26]. The study was limited by the lack of a control group.

Kriek et al., later conducted a study in 2017 to investigate whether alternate modes of SCS could help regain therapeutic benefits in patients who experienced a loss of analgesia over time [27]. This multicenter, double-blind, randomized and placebo-controlled study implanted 40 patients with refractory CRPS with an SCS device programmed to various settings. The following programmed settings included standard 40 Hz, 500 Hz, 1200 Hz, burst and placebo stimulation [27]. During a 10-week crossover period, the different SCS settings were programmed in a random order, with each setting being applied for a duration of two weeks. Scores were obtained using the visual analogue scale (VAS), McGill Pain Questionnaire (MPQ), and the Global Perceived Effect (GPE). At the conclusion of the crossover period, patients were given the opportunity to select the SCS setting they preferred. The authors found that the four different SCS settings resulted in significant pain reduction and higher patient satisfaction compared to placebo stimulation. These settings were all found to be equally effective, with 48% of patients preferring standard stimulation and 52% preferring non-standard stimulation [27]. Factors beyond pain relief may have played a role in their selection of preferred SCS setting, including the user-friendliness and comfort of each device. In summary, these studies demonstrated the effectiveness of SCS in alleviating pain and enhancing patient satisfaction.

## 6. Epidurals

Epidural infusion of opiates and local anesthetics has been studied as a treatment option for CRPS patients who have failed conservative treatments including physical therapy and medical management, however, success rates have been variable. In a retrospective study by Moufawad et al., 38 patients with CPRS resistant to conservative treatment, such as physical therapy and sympathetic blockade, received tunneled epidural catheters for a continuous infusion of fentanyl and bupivacaine, titrated to each patient over their initial and re-evaluation visits. The study identified factors associated with a higher pain control success rate, including pain confined to one limb, longer treatment length (eight weeks), and early initiation of the continuous epidural infusion (within the first year of symptom onset) [28].

Rauck et al., investigated the use of epidural clonidine for CRPS through a prospective, randomized, placebo-controlled study. This small trial recruited 26 patients with upper- or lower-limb CRPS resistant to sympathetic blocks. Patients were randomized to either low-dose clonidine, high-dose clonidine, or saline through cervical epidural catheters for upper-limb CRPS or lumbar epidural catheters for lower-limb CRPS. Pain scores were significantly improved in both clonidine groups compared to placebo six hours post-treatment. The low-dose clonidine group experienced less sedation compared to the high-dose group without significant differences in pain or hemodynamic changes [29]. However, the exact magnitude of the improvement was not reported, and it was not clear whether pain relief persisted for more than six hours after treatment. 

While the epidural infusion of opiates, local anesthetics, and alpha-2 adrenoceptor agonists may reduce the pain of CRPS, neuraxial control of CRPS should be used cautiously due to potential side effects such as hypotension and sedation. The risks of repeat epidural injections or tunneled epidural catheters are remarkably rare as the interventional pain field continues to advance patient safety but can have devastating effects if not performed by an experienced interventional pain specialist, including bleeding, infection, endocrine complications, and neurologic injury.

## 7. Chemical and Mechanical Sympathectomy

Many neuropathic pain syndromes, including CRPS, are thought to be mediated by a dysfunctional sympathetic nervous system. Historically, there have been attempts to interrupt sympathetically mediated pain including temporary and nondestructive measures such as local anesthetics, botulinum toxin, alcohol, and phenol injections to destroy sympathetic ganglia. Surgical ablation via the open removal or electrocoagulation of the sympathetic chain or stereotactic thermal/laser interruption has also been studied for neuropathic pain.

Prior reviews concluded that sympathectomies should be reserved for patients with severe CRPS refractory to other treatment modalities. Currently, surgical sympathectomies are much less commonly performed compared to percutaneous trials of local anesthetic or radiofrequency nerve ablation. One systematic review reported no significant difference between surgical and chemical sympathectomy for neuropathic pain relief; furthermore, the authors stated that the practice is based on weak evidence with potentially significant complications and should be used cautiously in clinical practice. 

In their systematic review and meta-analysis, Straube et al., aimed to assess the effectiveness of various sympathectomy techniques in reducing pain and improving function in patients with CRPS. The study included 11 randomized controlled trials (RCTs) with a total of 404 participants. One of the main findings of the study was that there was limited evidence to suggest that radiofrequency sympathectomy and neurolytic sympathectomy with phenol were equally effective in reducing pain and improving function in patients with CRPS [30]. Both techniques appeared to provide some benefit, although the evidence was not strong enough to conclusively recommend one technique over the other. However, the authors did note that there were some limitations to the available evidence, including a lack of standardization in the techniques used and variability in the patient populations studied [30]. Additionally, the quality of the studies included in the review varied, and there was a risk of bias in some of the trials. Overall, the study highlights the need for more high-quality research on the effectiveness of different sympathectomy techniques for CRPS. While evidence suggests that both radiofrequency sympathectomy and neurolytic sympathectomy with phenol may be effective, further research is needed to better understand the optimal technique and patient selection for this treatment approach.

## 8. Intrathecal Baclofen

Many patients with CRPS exhibit motor dysfunction with signs and symptoms including tremor, weakness, decreased range of motion, and dystonia. Dystonia is a comorbidity often associated with CRPS, affecting approximately 20% of patients. Marked by abnormal involuntary muscle contractions, predominantly fixed flexion postures, dystonia in CRPS has delayed onset and is often refractory to treatment, adding to the severe disease burden of CRPS. Loss of spinal GABAergic inhibition has been posited as an important mechanism in this type of dystonia. 

Intrathecal baclofen (ITB) has been studied for the management of CRPS-associated dystonia. Baclofen infused intrathecally around the spinal cord can stimulate presynaptic and postsynaptic GABA-b receptors. In 2009, van Rijn et al., found ITB delivered via pump for continuous administration reduces dystonia in CRPS over one year but was also associated with a high complication rate [31]. Adverse effects included symptoms of baclofen intoxication including nausea and vomiting. Catheter-related complications were common, including post-dural puncture headache [31]. Overall, the study suggests that ITB therapy may be an effective treatment option for dystonia in patients with CRPS, but careful monitoring is necessary to manage potential complications. 

In a later open-label study in 2013, van der Plas et al., investigated the effect of ITB on different pain qualities in 42 CRPS patients with dystonia. To study this effect, the group used the neuropathic pain scale (NPS), which includes 10 qualities of pain (intense, unpleasant, sharp, hot, dull, cold, sensitive, itchy, deep, and surface). The scale uses a numeric rating ranging from 0 (pain absent) to 10 (most severe pain) to quantify each pain quality item. Scores were evaluated every 3 months for 12 months. The study found that ITB resulted in a decrease in reported pain scores for pain that was qualified as intense pain, sharp pain, dull pain, and deep pain over the first 6 months of treatment [32]. After the first 6 months, reported pain scores leveled off despite the continually increased dose of ITB [32]. This study highlighted the benefit of ITB for pain reduction in patients with specific pain qualities in CRPS. This study was limited by the lack of a control group and lack of homogeneity in dose escalation of ITB.

## 9. Dorsal Root Ganglion Stimulation

Dorsal root ganglion stimulation (DRGS) appears to be a promising treatment option for CRPS, as it has shown similar success rates compared to spinal cord stimulation in reducing pain and improving mood. Over the past decade, there has been an increasing emphasis on electrical stimulation targeting the dorsal root ganglion, offering potential advantages in terms of improved target control. One significant factor is the cerebrospinal fluid layer that surrounds the DRG, which has a considerably smaller volume compared to the layer surrounding the spinal cord. As a result, DRG stimulation necessitates lower stimulation amplitudes compared to traditional SCS [33]. The “ACCURATE” study was the first randomized, controlled multicenter trial that compared DRGS to SCS in the setting of chronic pain of the lower limbs, secondary to CRPS [34]. In 2017, Deer et al., compared these two methods in 146 subjects with CRPS and found that DRGS resulted in greater success rates (over 50% reduction in pain score compared with baseline) compared to SCS at 3, 6, 9, and 12 months [35]. At 12 months, DRGS patients reported less paresthesia and experienced greater improvements in mood. However, no group differences were detected regarding patient satisfaction and serious adverse events.

A subgroup analysis was later conducted by Mekhail et al., in 2020, building upon the previously published ACCURATE study. They retrospectively analyzed 61 patients who had received a DRG neurostimulator implant. The study compared the outcomes of patients who achieved paresthesia-free stimulation to those who experienced paresthesia and measurements were taken at 1-, 3-, 6-, 9-, and 12-months follow-up. The percentage of patients who experienced pain relief without paresthesia increased from 16.4% at 1 month to 38.3% at 12 months [34]. The authors concluded that paresthesia-based neurostimulation was unnecessary for pain relief.

Huygen et al., also performed a comprehensive analysis by combining data from multiple published, prospective studies. Their aim was to identify differences in the effectiveness of DRG stimulation based on the underlying cause or location of pain. Additionally, they sought to investigate the generalizability and reproducibility of individual studies that tracked patients for a minimum of 12 months [33]. The pooled analysis revealed promising findings, demonstrating high responder rates and the effectiveness of DRG neurostimulation across various pain etiologies, with a particular focus on CRPS-I and CRPS-II. These outcomes align with independent retrospective studies that have evaluated DRG stimulation in chronic pain disorders such as phantom limb pain and chronic pelvic pain. Specifically, patients with CRPS-I and CRPS-II experienced substantial reductions in pain scores, with a 64% and 58% decrease, respectively, at the 12-month follow-up [33].

These findings provide robust evidence supporting the efficacy of DRG stimulation and highlight the potential of DRGS as a valuable treatment option for CRPS, but further research is still needed to fully understand the role of DRGS in the treatment of neuropathic pain.

## 10. Trigger/Tender Point Injections

A trigger or tender point is a knot or tight band of muscle that causes pain and tenderness. Trigger point injections (TPIs) are a treatment option that can be used to manage trigger points associated with CRPS, especially with the involvement of the upper limbs, trapezius, and suprascapular muscles. TPIs involve needle cannulation of the taut connective tissues and often injecting a small amount of local anesthetic and/or botulinum toxin into the trigger point. In the case of CRPS, trigger points can develop in the affected limb due to muscle tension and spasms. The goal of TPIs is to provide immediate pain relief and reduce inflammation in the trigger point. While there may be a small degree of muscle soreness post-injection, patients should remain active with a full range of motion, especially in the injected area(s), to reduce the recurrence of trigger point pathology. 

At the time of this writing, there is a paucity of research involving the use of TPIs in CRPS; however, research on TPI in the use of myofascial pain exists, which could provide insight into the role of TPI in CRPS. In 2009, a study compared trigger point injections with 1% lidocaine versus dry-needling, cannulation without any injection, in a study population with myofascial pain [36]. They found that both groups experienced a reduction in pain, cervical range of motion, and depression after 4 and 12 weeks, but no differences were found between the groups. Also in 2009, de Abreu Venancio et al., compared trigger point injections with 0.25% lidocaine versus 0.25% lidocaine and botulinum toxin versus dry needling, with a focus on headache management. While all three groups showed significant improvement, the Botox group experienced a significant reduction in the use of headache rescue medication and localized post-injection sensitivity.

A more recent study conducted by Su et al., provided a meta-analysis on the use of botulinum toxin injection via intramuscular injection, subcutaneous or intradermal injection, and lumbar sympathetic block in the management of CRPS. There was a significant reduction in pain at the first follow-up between three weeks and one month, but not at the second follow-up between two and three months [37]. Between the small number of trials that was included, eight, the heterogeneity of each, and the predominance of lumbar sympathetic blocks, we were unable to extrapolate clinically significant results specifically for TPI. 

There is a dearth of literature surrounding the use of TPI and CRPS. The use of TPI in patients with CRPS may be beneficial in the presence of trigger points, especially involving the shoulder girdles. There is no evidence-based consensus on what, if any, substance should be injected with a TPI. We recommend that patients with a component of trigger points to their CRPS trial TPIs with botulinum toxin and local anesthetic for pain-management optimization. 

## 11. Conclusions

Treating CRPS poses a formidable challenge, necessitating a multifaceted approach to address the numerous symptoms and underlying mechanisms involved. Throughout this article, we explored various interventional treatments aiming to manage CRPS. It is important to recognize that each intervention carries its own set of benefits and risks. From pharmacological approaches to physical therapies and invasive procedures, the selection of treatment should be tailored to the individual patient based on their unique circumstances and response to prior therapies. While some interventions may primarily target pain relief, others focus on improving functionality and enhancing the overall quality of life for CRPS patients. Balancing the potential benefits with the risks and potential side effects is crucial in ensuring the best possible outcomes. Furthermore, it is essential to emphasize the significance of a multidisciplinary approach in managing CRPS. Collaboration among healthcare professionals, including pain specialists, physiotherapists, psychologists, and occupational therapists, can optimize treatment plans and address the diverse aspects of the condition. Given the complex nature of CRPS, ongoing research and advancements in interventional treatments are necessary. Continual efforts to explore innovative approaches, refine existing techniques, and further understand the underlying mechanisms of CRPS will be instrumental in improving the outcomes for patients. By adopting a comprehensive and collaborative approach, healthcare providers can strive to enhance the management of CRPS and improve the lives of those affected by this complex disease.

## Data Availability

The authors confirm that the data supporting the findings of this study are available within the article.

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
