# Peer review of "Interventional Treatment of Complex Regional Pain Syndrome"

_biomedicines, 2023, doi:10.3390/biomedicines11082263_

Round 1

Reviewer 1 Report

There are several criticisms in this paper, which deserves substantial revisions.

Major revisions are reported below, which should be addressed and fixed by the Authors.

One major point is related to the fact that the present paper, which is intended to be a review article, contains very few cited literature references. Only 18 references for a review paper are not enough. As a review article, the paper should present current body of knowledge about the specific topic while drawing connections between research studies that have previously been published. Some references are very old, while some of the most recent ones are conflicting.

Furthermore, a review article should be accurate, comprehensive, and rigorous. In the paper the Authors refer to CRPS, but its description is poor and quite general.

With reference to the Budapest criteria and the IASP criteria, the Authors should be more precise, It is also fundamental to correct both in the abstract and introduction the term “chronic regional pain syndrome” since the term “chronic” is incorrect.

Since International Association for the Study of Pain (IASP) made up, in 1994 in Orlando, Florida, a task force on taxonomy to standardize the nomenclature into complex regional pain syndrome (CRPS), since till that time this disorder has been known by many names (such as reflex sympathetic dystrophy, Sudeck's atrophy, neurodystrophy, and so on); while Budapest criteria refer to CRPS symptomatology and differential diagnosis. So IASP and Budapest criteria refer to different aspect of this disorder.

The Authors should describe more in depth this painful and disabling condition known as CRPS type I, which instead is what they are referring to specifically when using the general term of CRPS. In fact, CRPS occurs in two types, namely CRPS type 1, or algodystrophy, and CRPS type II, once referred to as causalgia. These two types of disabling conditions are clearly distinguishable since CRPS type 1, which occurs without preceding nerve injury, while CRPS type 2 has the same clinical features as type 1, except for the presence of clinical signs and history consistent with a nerve injury (see https://doi.org/10.3390/app12188979). Furthermore, given its different clinical manifestations, CRPS has been divided by many authors into three distinct phases, which are not necessarily progressive, in relation to the time elapsed from the manifestation of signs and symptoms to diagnosis. This should be also discussed by the Authors.

Another important point that the Authors should take into account in order to revise the paper is tha fact that the types of interventions reported in the paper are not those intended as “conventional therapies” for CRPS treatment and also, they are mostly intended for CRPS type II.

The Authors should discuss more in detail on the role of physical, conventional, therapies (lines 47-49), also citing appropriate literature. Suggested pain management approaches comprise mirror therapy for brain functional reprogramming, physical therapy with transcutaneous electrical nerve stimulation (TENS), pain desensitization techniques, and the pulsating electromagnetic field technique (see https://doi.org/10.3390/app12188979).

In addition, these kind of interventions in Italy are not generally used in primary care, and in Europe they are rarely administered.

It is also important to highlight that most these interventions are generally administered in patients who have failed treatment with multiple drugs and procedures and/or are refractory to treatment (as also reported in the paper at lines 203-205).

Most of the treatment options reported in the text are mainly intended to obtain pain relief. However, these approaches should be applied only when traditional pain-relieving strategies have failed and If pain therapy has not yielded any improvement in the patient's quality of life, then it is possible to take into considerations these options.

Again, some of the procedures reported in the paper are based on very little high-quality evidence. So they should be used cautiously in clinical practice, in carefully selected patients, and who cannot bear physical programs due to intense pain, and again only after failure of other treatment options.

More high-quality evidence from double blind RCTs with placebo (sham) comparators is needed to determine the efficacy and safety of these procedures. Although improvement had been reported following the administration of steroids, epidural clonidine, intrathecal baclofen, spinal cord stimulation (SCS), and motor imagery programs, further trials are required to confirm these findings. In fact, as reported by the Authors “careful consideration of the benefits and risks should be taken into account for each individual patient.”, “careful monitoring is necessary to manage potential complications” and further research is needed to better understand the optimal technique and patient selection for this treatment approach.

Within this frame, the title of the paper should be revised. The use of the term “management” should be avoided; also “a Review of Safety and Efficacy” does not sound quite appropriate for this kind of article, since in each paragraph the Authors state that there are some limitations reported in the cited studies. Maybe the term “medical treatment of CRPS” sounds more appropriate.

Please also revise the conclusion section, since it is quite similar to the abstract and introduction section, while it should instead provide a summary of major agreements and disagreements in the literature and a summary of the general conclusions drawn, as well as any gaps or areas for further research and/or overall perspective on the topic.

Finally, another critical point is that it seems that each paragraph is separated/independent from the previous and the following ones, and this makes the reading not so fluent. The Authors should revise the text to implement the readability of the paper and overall, also the subdivision into paragraphs.

Finally, with reference to paragraph “2. Multidisciplinary Approaches to Treatment”, it this paragraph necessary or could this section being integrated in the introduction? If not, this aspect should be strengthened also in the following paragraphs as well as in the conclusion section.

The english language used is not sufficiently comprehensible throughout all the manuscript and can be improved and some sentences can be more fluent.

I suggest the authors get editing help from someone with full professional proficiency in English.

Author Response

Thank you for your feedback. Tho following changes have been made.

-Additional information has been added with newer resources while also addressing points made from older references. Both older and more recent articles were used to describe both the historical treatment of CRPS as well as new emerging information regarding the disease. While some of these article provide conflicting arguments for or against different modes of treatment, this article weighs in on the pros and cons of each treatment. 

-The Budapest and IASP criteria has been described with more detail. The term "Chronic" has also been removed from describing CRPS throughout the article. 

-The history of CRPS along with its changes in nomenclature and different types (CRPS 1 vs. 2) have been described in more detail in the beginning paragraphs of the article. 

-The role of other conventional therapies for CRPS have also been described in more detail. While the focus of this article is on the interventional treatments for CRPS, we have touched upon different medical treatments as well. It was also highlighted that the interventional treatments are reserved for patients with symptoms refractory to conventional treatment. 

-The title of the paper has been revised, with the term "management" replaced with treatment. This paper focuses on the interventional treatment of CRPS as opposed to solely medical treatment.

-The article has been revised in order to be more readable as one fluent text.

-The "Multidisciplinary Approaches to Treatment" section has been strengthened to describe the various treatment modalities that are often initiated in patients before attempting more invasive procedures. This section also touches upon the complex nature of the disease requiring professional care from various disciplines.

Reviewer 2 Report

This is a review of interventions in CRPS. The paper is well written and within its limitations it cites evidence appropriately. However, while the interventions component is broadly fine, the real excitement from such review articles comes from a paper that manages to teach and inspire clinicians to use the included interventions within real world clinical practice. Unfortunately, this paper falls a bit short of that. Specifically, I would like to see a little more nuance in the diagnosis and multidisciplinary approach. How does the anesthesiology intervention fit into a proper management plan? When is it appropriate? While I could be wrong, I really feel like the junior staff on this paper played a larger role in its development than the senior authors/clinicians.

The definition of CRPS and its diagnosis in the introduction first sentence is  severely lacking. CRPS is not just pain - although recognized in subsequent sentences this should be amended. The diagnostic criteria should be defined in a more nuanced fashion. Especially the trajectory of the IASP to Budapest criterion development and the difference between clinical and research criteria and when and why this matters. This is especially critical when we are assessing patients for invasive treatments that may cause iatrogenic harm (eg sympathotectomies). It might be ok to overdiagnose for physical therapy but should we be more strict for ketamine coma (i know this is not discussed but it is used)? These considerations should be a part of any clinician's decision making process.

Mechanisms of CRPS have been extensively explored and a lot is known from SNS dysfunction, to immune and neuroimmune changes, to brain plasticity changes. Many of these mechanisms point to specific treatments that have shown some utility. Yet, the authors do not discuss this literature and merely state that "the underlying mechanisms of CRPS are still not fully understood".

After using CRPS, why go back to calling it chronic regional pain syndrome? Also, the authors forgot to discuss RSD and Sudeck's and other common albeit archaic names. Chronic RPS is not something that is used frequently except when people mistakenly think that C stands not for complex but for chronic. This misnomer is important because CHRONIC implies that CRPS can only occur >12 weeks and yet we know that CRPS signs may develop very early. The other names I mentioned are more common although disappearing.

The multidisciplinary section just describes what is the case for all of medicine. Nothing unique to CRPS is described nor are any outcomes data provided for these approaches. See my other comments about a missed opportunity for contextualizing the interventional section.

Trigger points are not an area that most practitioners would recommend (or even believe in) for CRPS. I am confused about why these are included as part of CRPS management. Although the authors may disagree, see DOI: 10.1093/rheumatology/keu471 for an example of a criticism of these techniques. Granted that this is something that may be done in their practice however, why would a technique that is essentially not part of any evidence based practice guidelines be chosen to be presented as the first technique? It is fine to include with appropriate limitations but it really should be at the end of the line.

The rest of the article is about various iterations of injections. This is fine but to make this paper stand out and be more clinically useful, the authors should make an attempt to place injections within what is commonly called the biopsychosocial framework within which all modern pain management exists. While they mention multidisciplinary teams, their roles and potential therapies should be at least briefly explained and cited. I am thinking of CBT, mirror therapy, etc. All of which have "superior" outcomes to interventions (when I say superior, it is with the understanding that these are difficult to compare head to head as they have different reasons for implementation; but again, this subtlety in clinical decision making is not displayed and would make this vastly more interesting).  Interventions are critical BUT they must be placed within the context of a comprehensive management plan. I feel like the key part of such a paper should be to help the inexperienced practitioner understand WHEN to intervene and WHICH intervention to use based on existing scientific enquiry.

The authors may consider whether CRPSI vs II is appropriate to discuss somewhere.

Author Response

Thank you for your feedback. The following changes have been made in response:

A more detailed description of the history of CRPS, its changes in nomenclature, different types, and diagnosis has been added to provide a more complete background. The section regarding multidisciplinary and conventional treatments has been edited to provide more detail as well, including descriptions of physical, psychological, and conventional medical therapies for CRPS. It has been noted that each of these treatments have their own limitations and the benefit-risk ratio must be considered when treating patient with symptoms refractory to conventional treatment.

We have included more detailed description of the mechanisms behind CRPS, including inflammatory, immunologic, and brain plasticity changes. While the mechanisms of CRPS have been explored, it is mentioned that it is not yet fully understood and therefore remains a challenge to treat.

The term "chronic regional pain syndrome" has also been removed and the history of various nomenclature has been discussed in more detail. 

Round 2

Reviewer 2 Report

I'm happy with the responses. Some newer mechanistic articles and systematic review could have been included as part of the disease mechanisms discussion but it's OK. 

It could be improved for flow (eg not using the word criteria twice in a short sentence) but it is ok.